# A Comparison of McGrath Videolaryngoscope versus Macintosh Laryngoscope for Nasotracheal Intubation: A Systematic Review and Meta-Analysis

**DOI:** 10.3390/jcm11092499

**Published:** 2022-04-29

**Authors:** Chia-Hao Ho, Li-Chung Chen, Wen-Hao Hsu, Tzu-Yu Lin, Meng Lee, Cheng-Wei Lu

**Affiliations:** 1Department of Anesthesiology, Far Eastern Memorial Hospital, Banqiao Dist., New Taipei City 220, Taiwan; tad820423@gmail.com (C.-H.H.); burrn.down@gmail.com (L.-C.C.); dortan2011@gmail.com (W.-H.H.); drlinfemh@gmail.com (T.-Y.L.); 2Department of Mechanical Engineering, Yuan Ze University, Taoyuan 320, Taiwan; 3Department of Neurology, Chang Gung University College of Medicine, Chiayi Chang Gung Memorial Hospital, Puzi City, Chiayi 613, Taiwan

**Keywords:** McGrath, videolaryngoscope, nasotracheal intubation

## Abstract

Background: In this study, it was shown that the routine use of McGrath videolaryngoscopy may improve intubation success rates. The benefits to using a videolaryngoscope in nasotracheal intubation were also demonstrated. However, no solid evidence concerning the effectiveness of the use of McGrath videolaryngoscopes in nasotracheal intubation has previously been reported. As a result, we questioned whether, in adult patients who underwent oral and maxillofacial surgeries with nasotracheal intubation (P), the use of a McGrath videolaryngoscope (I) compared with a Macintosh laryngoscope (C) could reduce the intubation time, improve glottis visualization to a score of classification 1 in the Cormack–Lehane classification system, and improve the first-attempt success rate (O). The secondary outcomes measured were the rate of the use of Magill forceps and the external laryngeal pressure (BURP) maneuver used. Methods: An extensive literature search was conducted using databases. Only randomized controlled trials that compared the McGrath videolaryngoscopy and Macintosh laryngoscopy techniques in nasotracheal intubation in adult patients were included. Results: Five articles met the inclusion criteria and were included in the final analysis (*n* = 331 patients). The results showed a significant decrease in intubation time and a higher rate of classification 1 scores in the Cormack–Lehane classification system, but no difference in the first-attempt success rates were found between the McGrath group and the Macintosh group. Decreases in the rate of the use of Magill forceps and the use of the external laryngeal pressure maneuver were also found in the pooled analysis. With regard to the overall risk of bias, the selected trials were classified to have at least a moderate risk of bias, because none of the trials could blind the operator to the type of laryngoscope used. Conclusions: Our analysis suggests that the use of a McGrath videolaryngoscope in nasotracheal intubation resulted in shorter intubation times, improved views of the glottis and similar first-success rates in adult patients who received general anesthesia for dental, oral, maxillofacial, or head and neck cancer surgery, and also reduced the use of Magill forceps and the BURP maneuver.

## 1. Introduction

McGrath videolaryngoscopes comprise a direct video laryngoscope, a battery-contained handle, and a disposable plastic blade in a single device, and anesthesiologists can perform intubation using a McGrath videolaryngoscope in patients with either normal or difficult airways. Kriege et al. revealed that the routine use of McGrath videolaryngoscopy may improve intubation success rates [1].

Hoshijima et al. completed a comprehensive systematic review and meta-analysis in which a comparison of McGrath videolaryngoscopes versus Macintosh laryngoscopes in orotracheal intubation was presented. The authors suggested that the McGrath videolaryngoscope was more suitable than the Macintosh laryngoscope in terms of glottic visualization, but the McGrath videolaryngoscope extended the intubation time, and its success rate in terms of tracheal intubation was not superior [2]. However, the study did not compare these two tools in nasotracheal intubation.

Nasotracheal intubation is largely performed during oral and maxillofacial surgeries. The benefits of this technique include the fact that it provides good accessibility and a larger surgical field [3]. The procedure involves passing an endotracheal tube through the nostril into the nasopharynx and the trachea. Several techniques are often used to enhance the success rate during nasotracheal intubation, such as the use of Magill forceps or the external laryngeal pressure maneuver [3]. A systematic review and meta-analysis by Jiang et al. compared videolaryngoscopy with direct laryngoscopy in nasotracheal intubation and concluded that the use of a videolaryngoscope did not improve the success rate of nasotracheal intubation in adult patients, but it improved the first-attempt success rate, optimized the laryngeal view, and decreased the intubation time. Additionally, the analysis showed a lower rate of the use of Magill forceps [4]. However, studies that used videolaryngoscopy included in this meta-analysis were detailed, and this study did not specifically survey the McGrath videolaryngoscopy technique.

Currently, there is no solid evidence concerning the effectiveness of McGrath videolaryngoscopes in nasotracheal intubation. As a result, we questioned whether, in adult patients who underwent oral and maxillofacial surgeries with nasotracheal intubation (P), the use of a McGrath videolaryngoscope (I) compared to a Macintosh laryngoscope (C) could reduce intubation time, improve glottis visualization to a score of classification 1 in the Cormack–Lehane classification system, and improve the rate of first-attempt success in intubation (O). The secondary outcomes were the rate of the use of Magill forceps and the use of the external laryngeal pressure (BURP) maneuver.

## 2. Methods

This study followed the Cochrane Handbook for Systematic Review of Interventions [5] and the Preferred Reporting Items for Systematic Reviews and Meta-Analysis (PRISMA) [6], and the study protocol was registered in the International prospective register of systematic reviews (PROSPERO) in 2022 (registration number: CRD42022293199).

### 2.1. Search Strategy

An extensive literature search was conducted using PubMed and Cochrane Central Register of Controlled Trials from 1 January 1980 to 10 October 2021. The last search date was 1 November 2021. The search strategy used in the two electronic databases was the use of search strings, including “McGrath, (or McGrath MAC, videolaryngoscope), and nasotracheal intubation (or NTI)” in all fields. Reviews, case reports, and studies published in abstract forms were excluded. No language restriction was imposed.

### 2.2. Study Selection

1.Inclusion CriteriaProspective randomized clinical trials that compared the McGrath videolaryngoscopy and Macintosh laryngoscopy techniques in nasotracheal intubation in adult patients (age ≥ 18 years old) who underwent operations with general anesthesia were included.2.Exclusion CriteriaWe excluded manikin trials, cadaver studies, observational studies, studies that involved tracheal intubation during cardiopulmonary resuscitation, double-lumen tubes, pediatric patients (age < 18 years old), and articles that involved nasotracheal intubation with other videolaryngoscopes.

### 2.3. Outcomes

1.Primary outcomeThe primary outcomes were the intubation time (from the intranasal placement of the tube to the detection of carbon dioxide via capnography), the rate of classification 1 scores in the Cormack–Lehane classification system, and the first-attempt success rate.2.Secondary outcomeThe secondary outcomes were the rate of the use of Magill forceps and the use of the external laryngeal pressure (or backward, upward, or rightward pressure) maneuver.

### 2.4. Data Extraction

Three authors (Ho, CH, Hsu, WH, and Chen, LC) assessed each article independently, evaluated whether it met the inclusion criteria, and used standardized data collection forms for data extraction. For continuous variables, the mean, standard deviation (SD), and sample size were extracted from each eligible article. Data such as the median and interquartile range that could not be used directly were converted to means and SDs using formulae provided in the Cochrane Handbook. For the dichotomous data, the number of events that occurred and the sample size were also extracted. If more than two comparisons were made in one study, the authors only extracted the results concerning the McGrath videolaryngoscopy and Macintosh laryngoscopy groups.

### 2.5. Data Synthesis

In terms of the data synthesis of the outcomes involved in the studies, three types of outcomes were observed:All of the studies included shared the same methods and units when evaluating the outcomes, such as intubating time, first-attempt success rate, the Cormack–Lehane classification of the quality of the view of the glottis/vocal cord, and the use of Magill forceps during intubation (continuous outcomes needed to share the same unit);When evaluating the outcomes, different terms which shared one similar meaning were used: external laryngeal manipulation. Some of the studies used the term “backward–upward–rightward pressure maneuver (BURP maneuver)” or “external laryngeal pressure” to define the same maneuver.The studies included used different tools/values to evaluate the outcome of ease of intubation. This kind of outcome was not synthesized and included in our studies.

Furthermore, the data were only synthesized and evaluated when more than 50% of the studies had thoroughly included data-concerning outcomes.

Five groups of data met the criteria and were synthesized: intubation time, Cormack–Lehane grade, the use of Magill forceps, external laryngeal pressure, and first-attempt success rate. Meta-analyses were performed using RevMan 5.4 software, The Nordic Cochrane Centre, Copenhagen, Denmark (https://training.cochrane.org/online-learning/core-software-cochrane-reviews/revman/revman-5-download, accessed on 21 October 2021). A random-effects model was applied to account for clinical and methodological heterogeneity between studies. Statistical heterogeneity was assessed with I^2^, where values of 30–60% and 50–90% were considered to represent moderate and substantial heterogeneity, respectively. The risk ratios (RRs) or odds ratios (ORs) with 95% CIs were calculated for dichotomous/discrete outcomes and then pooled with the Mantel–Haenszel method. Continuous outcomes (intubation time) were calculated with the weighted mean differences (WMDs) of mean values and SDs using the inverse variance method. A *p*-value < 0.05 was considered statistically significant. The outcomes of intubation time and Cormack–Lehane grade were analyzed using a random-effects model, and the other three study data were analyzed using a fixed-effects model.

### 2.6. Risk of Bias

Two authors (Ho, CH and Hsu, WH) independently appraised the risk of bias of the selected eligible studies using the “risk of bias” assessment tool in the Cochrane Handbook and generated a “risk of bias” summary figure using Review Manager (RevMan 5.4.1). Concerning the overall risk-of-bias judgement, if the trial was assessed to be at low risk of bias in all domains for this result, the study was classified as being “low-risk”; if the trial was assessed to raise some concerns in more than one domain without any high risk of bias in any domain, the study was classified as having “some concerns”; if the trial was assessed to be at high risk of bias in more than one domain, the study was classified as being “high-risk”.

### 2.7. Quality Assessment

The quality of evidence concerning the outcomes that we investigated was assessed by applying the Grading of Recommendations Assessment, Development, and Evaluation (GRADE) system to study the limitations, consistency of effects, imprecision, indirectness, and publication bias in our reviews [7]. After the assessment, a table concerning the GRADE evidence profile was created using GRADEpro software (https://www.gradepro.org/, accessed on 18 April 2022) to rate all outcomes, including very low, low, moderate, or high quality.

## 3. Results

### 3.1. Searching Result

Following our search strategy, 67 papers were found on PubMed, and 387 papers were found on Cochrane. Duplicated and unpublished studies were excluded initially, and the remaining 236 studies were screened carefully using their titles and abstracts. A total of 224 studies were excluded at this step, of which 164 studies were irrelevant (including studies concerning orotracheal intubation or different topics), 7 studies discussed videolaryngoscopes, 2 studies were manikin studies, and 51 studies were not RCTs. In total, 12 articles were selected for full text assessment following our inclusion and exclusion criteria. A total of seven articles were excluded, of which one article discussed airways that were predicted to be difficult and did not have adequate outcomes that we could analyze [8], one article discussed a pediatric population [9], and five articles discussed different videolaryngoscopes [10,11,12,13,14]. Eventually, five articles were found to meet our inclusion criteria and were included in the final analysis (*n* = 331 patients) [15,16,17,18,19] (Figure 1).

### 3.2. Included Studies

The characteristics of the selected studies are summarized in Table 1. In the meta-analysis, a total of 331 cases were included (165 cases that used the McGrath laryngoscope and 166 cases that used the Macintosh laryngoscope). The type of surgery performed in the selected studies included dental, oral, maxillofacial, and head and neck cancer surgery. All of the participants were classified as ASA 1~2. All five studies were carried out in patients with normal airways.

### 3.3. Result of Primary Outcomes

In the analysis of the five selected studies, the results showed significant decreases in intubation times in the McGrath group compared with the Macintosh group (MD, −10.98 sec; 95% CI, −18.97 to −2.98; *n* = 331; *p* = 0.007; I^2^ = 88%, Figure 2). During intubation, when using McGrath videolaryngoscope, there was a greater possibility of obtaining a view of the vocal cords that was classified as classification 1 in the Cormack–Lehane classification system (RR, 2.34; 95% CI, 1.25 to 4.40; *n* = 331; *p* = 0.008; I^2^ = 87%, Figure 3), which indicated that using the McGrath videolaryngoscope provided better glottis visualization during nasotracheal intubation. All of the trials separately revealed significantly better Cormack–Lehane classifications when the McGrath videolaryngoscope was used. Pooled data showed no significant differences in the first-attempt success rates between the McGrath and Macintosh laryngoscopes (RR, 1.04; 95% CI, 1.00 to 1.08; *n* = 331; *p* = 0.17; I^2^ = 38%, Figure 4).

### 3.4. Result of Secondary Outcomes

All five studies reported a comparison of the rate of the use of Magill forceps. The pooled analysis showed that McGrath videolaryngoscopy compared with Macintosh laryngoscopy was associated with a reduced rate of Magill forceps use (OR, 0.08; 95% CI, 0.03 to 0.23; *n* = 331; *p* < 0.00001; I^2^ = 0%, Figure 5). The use of the external laryngeal pressure maneuver was compared in four studies. Kwak et al. used optimal external laryngeal manipulation despite what the Cormack–Lehane classification was and compared the quality of glottis visualization before and after optimal external laryngeal manipulation. The pooled analysis revealed a significant difference between the two groups (OR, 0.13; 95% CI, 0.07 to 0.25; *n* = 261; *p* = 0.002; I^2^ = 80%, Figure 6).

### 3.5. Risk of Bias

The risks of bias are summarized in Figure 7, and the overall risk of bias in the selected trials was classified to be at least a moderate risk of bias, because during all of the trials, blinding of the type of laryngoscope to the participants is impossible. In addition, Chae et al. did not present adequate outcomes that we could include.

### 3.6. GRADE Assessment

We evaluated the quality of evidence according to the GRADE assessment [7], and Table 2 displays a brief summary of the quality of evidence and the findings. Due to heterogeneity and the impossibility of blinding the participants, most of the outcomes were rated as low to very low quality, which is one of the limitations of our study.

## 4. Discussion

Our analysis showed that the use of the McGrath videolaryngoscope shortens the nasotracheal intubation time compared with that needed using the Macintosh laryngoscope. This result was compatible with that of the previous meta-analysis [4]. Jiang et al. reported a pooled analysis that showed shorter intubation times in nasotracheal intubation using different videolaryngoscopes. However, Hoshijima et al. reported a prolonged orotracheal intubation time when the McGrath videolaryngoscope was used compared with when the Macintosh laryngoscope was used [2]. The difference in these results may be the result of differences between the process of nasotracheal intubation and traditional orotracheal intubation. The process of nasotracheal intubation includes passing an endotracheal tube through the naris into the nasopharynx and using the laryngoscope to visualize the endotracheal tube that passes through the vocal cords. Operators were not able to adjust the shape of the endotracheal tubes using stylets. Due to the process and limitations mentioned above, the time needed for nasotracheal intubation is more unpredictable and relies more heavily on the view of the glottis using a laryngoscope. Furthermore, most of the participants in our studies underwent dental, oral, maxillofacial, and head and neck cancer surgery, which means our population was very different from that studied by Hoshijima et al. Concerning the comparison of the McGrath videolaryngoscope and other videolaryngoscopes, the McGrath videolaryngoscope allowed for a shorter orotracheal intubation time as the patients had restricted neck movement and were limited in their ability to open their mouths [20].

The results of our study also display an increase in the rate of classification 1 scores from the Cormack–Lehane classification system when using the McGrath videolaryngoscope, which suggests that the glottis can be visualized better using this technique. A previous study revealed that laryngeal grade views were superior to the McGrath videolaryngoscope than the Macintosh laryngoscope in simulated difficult airways [21]. The improvement of glottic visualization provided by videolaryngoscopes may be attributed to the digital camera on the blade tip of videolaryngoscopes, which allows practitioners to access the glottis more intuitively, gain a wider visual angle, and decrease the demand of the alignment of the visual axes. In order to predict the rate of difficult intubation, the Cormack–Lehane classification system was utilized to describe the views of laryngeal structures via direct laryngoscopy. Nevertheless, it was questioned whether this classification was appropriate for predicting the success rate with videolaryngoscopy [22]. Videolaryngoscopy provides indirect views of the glottis, so practitioners should have good hand–eye coordination and the adequate experience required to perform videolaryngoscopies.

A previous study showed that the results concerning the first-attempt success rates between McGrath and Macintosh laryngoscopy in tracheal intubation were similar, and first-attempt success rates were only increased in patients with difficult airways using videolaryngoscopy in nasotracheal intubation [5]. The results of our study revealed a similar result that showed McGrath and Macintosh laryngoscopes were not statistically different in terms of the first-attempt success rates. However, a previous study that investigated the use of these techniques in patients with predicted difficult airways showed that the use of the McGrath videolaryngoscope increased the first-attempt success rate [8]. 

In the nasotracheal intubation procedure, practitioners often use assistive maneuvers to pass the endotracheal tube through the vocal cords, including Magill forceps, the BURP maneuver, cuff inflation, etc. In our analysis, the rates of the use of Magill forceps and external laryngeal pressure were much lower in nasotracheal intubation procedures that utilized the McGrath videolaryngoscope. Previous RCTs demonstrated the same conclusion that using videolaryngoscopes in nasotracheal intubation resulted in fewer uses of Magill forceps compared with using a conventional direct laryngoscope [23]. Fewer uses of assistive maneuvers could not only represent clearer laryngeal views but also reduce possible complications, such as direct pharyngeal injury and cuff tear [24,25]. While nasotracheal intubation is associated with numerous complications [26], anesthesiologists should be concerned about every possible complication.

There were several limitations in our analysis. First, every study that we included had a different study protocol, strategy, and endpoint, which meant that measurement biases on primary and secondary outcomes were present in our analysis. Second, all of the participants that we enrolled were adults; therefore, these results cannot be directly applied to pediatric populations. Additionally, one trial gave the contradictory suggestion that the Macintosh laryngoscope provided shorter nasotracheal intubation times, better tracheal navigation, and required less use of the cuff inflation method in a pediatric population [9]. Third, nasotracheal intubation was usually performed in patients with predicted difficult airways; however, the cases in our analysis were classified as normal airways. As a result, the conclusions would be difficult to apply to the case of predicted difficult airways. Finally, two factors decrease the quality of the evidence of our outcomes: one is the impossibility of blinding due to the different appearance of the two intubating tools, and the other is the heterogeneity of the studies included.

In conclusion, our analysis suggests that using the McGrath videolaryngoscope in nasotracheal intubation provided shorter intubation times, better glottis views, and higher first-success rates in adult patients who received general anesthesia for dental, oral, maxillofacial, or head and neck cancer surgery, and also reduced the uses of Magill forceps and the BURP maneuver. However, additional high-quality trials should be obtained to clarify the benefits of the McGrath videolaryngoscope in terms of the overall success rate, in pediatric populations and in predicted difficult airways.

## Figures and Tables

**Figure 1 jcm-11-02499-f001:**
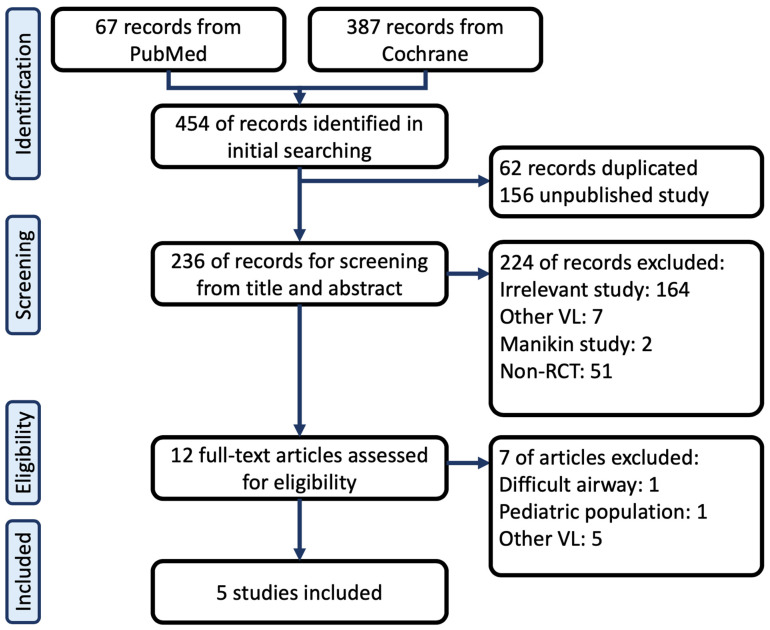
Flow chart of the systemic review. VL, videolaryngoscope; RCT, randomized control trial. Articles Excluded due to Difficult airway [8], Pediatric population [9], Other VL [10,11,12,13,14]; Included studies [15,16,17,18,19].

**Figure 2 jcm-11-02499-f002:**
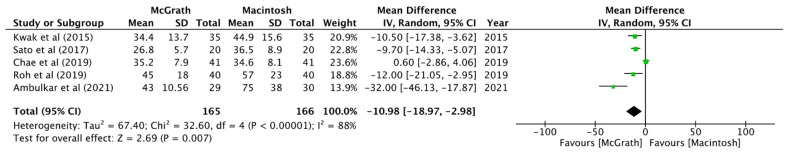
Forest plot of the intubation time of nasotracheal intubation (McGrath vs. Macintosh laryngoscope). The width of the horizontal line represents the 95% confidence interval (CI) of each study, and the square proportional represents the weight of each study. The rhombus represents the pooled rate and 95% CI. (same as below).

**Figure 3 jcm-11-02499-f003:**
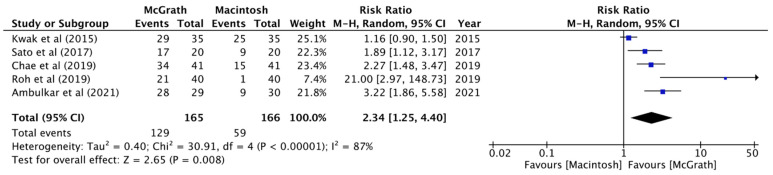
Forest plot of the rate of Cormack–Lehane classification 1 (McGrath vs. Macintosh laryngoscope).

**Figure 4 jcm-11-02499-f004:**
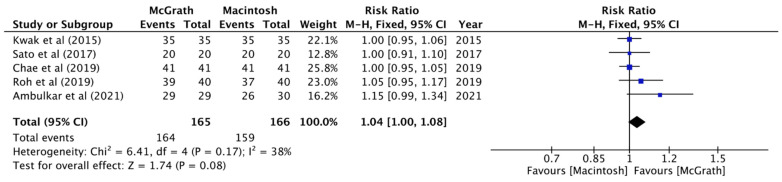
Forest plot of the first-attempt success rate (McGrath vs. Macintosh laryngoscope).

**Figure 5 jcm-11-02499-f005:**
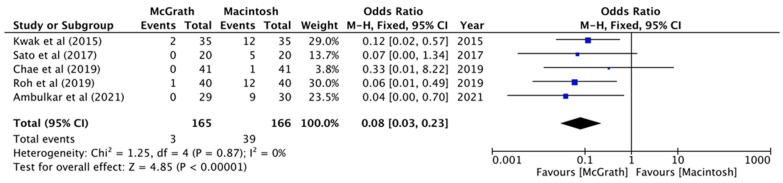
Forest plot of the Magill forceps used (McGrath vs. Macintosh laryngoscope).

**Figure 6 jcm-11-02499-f006:**
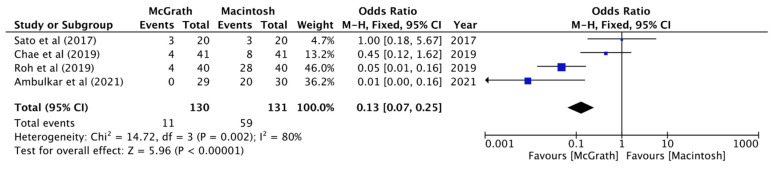
Forest plot of the external laryngeal pressure maneuver used (McGrath vs. Macintosh laryngoscope).

**Figure 7 jcm-11-02499-f007:**
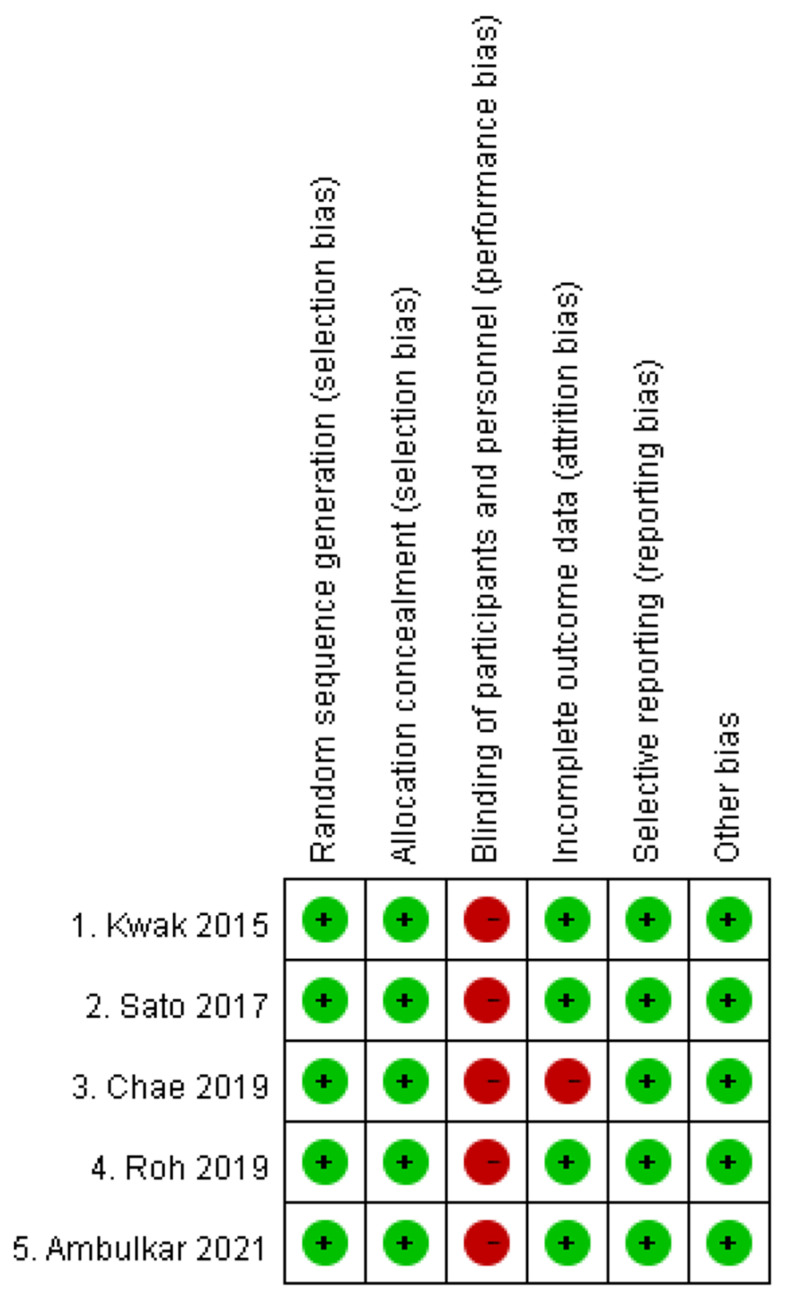
The summary of the risks of bias of the selected studies. Green circle with a plus symbol represents low risk of bias, and red circle with a minus symbol represents high risk of bias. All the studies face high risk of bias regarding the blinding of the participants.

**Table 1 jcm-11-02499-t001:** The summary of the characteristics of included studies.

Author	Year	Participants	Case Number (MG/ML)	ASA Status	Outcomes					Other Outcomes
					Intubation Time	CL Classification 1	Successful Rate in 1st Attempt	Magill Forceps Use	BURP Maneuver	
Kwak [15]	2015	Oral and maxillofacial surgery	70 (35/35)	1~2	V	V	V	V		Ease of intubation, bleeding
Sato [16]	2017	Elective oral surgery	40 (20/20)	1~2	V	V	V	V	V	bleeding, esophageal intubation, dental injury
Chae [17]	2019	Elective oral and maxillofacial surgery	82 (41/41)	1~2	V	V	V	V	V	Nasotracheal intubation difficulty score
Roh [18]	2019	Dental or maxillofacial surgery	80 (40/40)	1~2	V	V	V	V	V	Bleeding risk, ease of intubaiton
Ambulkar [19]	2021	Elective head and neck cancer surgery	59 (29/30)	1~2	V	V	V	V	V	Difficulty of intubation

MG: McGrath laryngoscope, ML: Macintosh laryngoscope, ASA status: American Society of Anesthesiologists Classification, CL: Cormack–Lehane classification; BURP: backward, upward, right lateral pressure maneuver.

**Table 2 jcm-11-02499-t002:** GRADE Evidence Profiles: McGrath for nasotracheal intubation.

Quality Assessment						Summary of Findings				
No. of Studies	Risk of Bias	Inconsistency	Indirectness	Imprecision	Publication Bias	Number of Patients (%)		Effect		Quality of Evidence
						McGrath	Macintosh	Relative Risk (95% CI)	Absolute Risk	
Intubation time										
5	No serious risk of bias *^1^	Serious *^2^	No serious limitation	No serious limitation	No Serious limitation	165	166	MD = −10.98 (−2.98~−18.97)		Low
Cormack-Lehane classification 1										
5	No serious risk of bias *^1^	Serious *^2^	No serious limitation	No serious limitation	No Serious limitation	129/165 (78.2%)	59/166 (35.5)	2.34 (1.25~4.40)	44 more per 100	Very low
First attempt successful rate										
5	No serious risk of bias *^1^	No serious limitation	No serious limitation	Mild limitation	No Serious limitation	164/165 (99.4%)	159/166 (95.8%)	1.04 (1.00~1.08)	Not Significant	Low
Use of Magill forceps										
5	No serious risk of bias *^1^	No serious limitation	No serious limitation	No serious limitation	No Serious limitation	3/165 (1.8%)	39/166 (23.5%)	OR = 0.08 (0.03–0.23)	21 less per 100	Low
Backward-upward-rightward Pressure Maneuver										
4	No serious risk of bias *^1^	Serious *^2^	No serious limitation	No serious limitation	No Serious limitation	11/130	59/131	OR = 0.13 (0.07–0.25)	36 less per 100	Very low

MD: mean difference, OR: odds ratio. *^1^: All the trials involved has the risk of bias due to incapability of blinding of the participants; *^2^: substantial heterogeneity found (I^2^ between 60–90%). Quality of evidence: low means that confidence in the effects of the intervention is very likely to change with future research findings or all studies have severe limitations; very low means that uncertainty remains about the effects of the intervention.

## Data Availability

Not applicable.

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
