# Peer review of "A Comparison of McGrath Videolaryngoscope versus Macintosh Laryngoscope for Nasotracheal Intubation: A Systematic Review and Meta-Analysis"

_jcm, 2022, doi:10.3390/jcm11092499_

Round 1

Reviewer 1 Report

Thank you for submitting this systematic review comparing a video laryngoscope with the direct laryngoscope for nasotracheal intubations. I have read this manuscript with interest.

Here are a couple of comments and suggestions:

Page 2 line 44-45: COVID has little to do with this systematic review. This sentence, including reference 2 should be excluded.

Page 3 line 108-112: Could the authors explain the rationale for the stated criteria for data synthesis?

Page 3 line 119-120: “Statistical heterogeneity was assessed with I², where values of 30 - 60% and 50 - 119 90% were considered as moderate and substantial heterogeneity, respectively.” There is an overlap for 50-60%. Do the authors have a reference for this?

Page 3 line 127-128: “If at least 10 trials are identified, we will use 127 funnel plots to determine publication bias.” This sentence can be excluded since the authors only included 6 studies.

Page 5 Table 1: Reference numbers can be indicated for the 6 papers.

Page 6 Figure 3: Why did the authors choose to use odds ratio for Cormach and Lehane (C&L) grade 1 comparing the groups? The incidence of C&L grade 1 is noted to be high. Why not use relative risk?

Page 8 line 216-217: “Application of stylet to 216 adjust the shape of the endotracheal tube is incapable.” Please rephrase this sentence.

Page 8 line 240-241: Although 1st attempt success rate was statistically significant between the groups, the clinical significance is less obvious with an OR of 1.06. This should be highlighted to the readers. If the study by Zhu et al is excluded, I suspect the statistical significance would also no longer be present.

Page 8 line 257-259: “Furthermore, one trial gave the opposite suggestions that Macintosh 257 laryngoscope provided shorter nasotracheal intubation times, better tracheal navigation, 258 requiring less use of the cuff inflation method.” Which study is this? The reference should be given.

Author Response

We thank the reviewer for the critical comments and constructive suggestions.

Page 2 line 44-45: COVID has little to do with this systematic review. This sentence, including reference 2 should be excluded.

Response 1:

We have deleted the reference that was related COVID in our introduction because COVID-19 was an irrelevant topic.

Page 3 line 108-112: Could the authors explain the rationale for the stated criteria for data synthesis?

Response 2:

We have added the explanation of the rationale for the stated criteria for data synthesis (Page 3 line 119-134).

Page 3 line 119-120: “Statistical heterogeneity was assessed with I², where values of 30 - 60% and 50 - 119 90% were considered as moderate and substantial heterogeneity, respectively.” There is an overlap for 50-60%. Do the authors have a reference for this?

Response 3:

The overlap is due to the fact that several factors may affect the importance of inconsistency and thresholds for the interpretation of I2 can be misleading. The above range is referred from https://training.cochrane.org/handbook/current

10.10.2 Identifying and measuring heterogeneity

Page 3 line 127-128: “If at least 10 trials are identified, we will use 127 funnel plots to determine publication bias.” This sentence can be excluded since the authors only included 6 studies.

Response 4:

As suggestion by the reviewer, the sentence has been excluded.

Page 5 Table 1: Reference numbers can be indicated for the 6 papers.

Response 5:

As suggestion by the reviewer, reference numbers were indicated, and Zhu et al was excluded by another reviewer’s suggestion.

Page 6 Figure 3: Why did the authors choose to use odds ratio for Cormach and Lehane (C&L) grade 1 comparing the groups? The incidence of C&L grade 1 is noted to be high. Why not use relative risk?

Response 6:

As suggestion by the reviewer, we used relative risk to analyze the outcome Cormack-Lehane grade (C&L) grade 1 comparing the groups. And we also agreed that risk ratio was more appropriate method to analyze this outcome.

Page 8 line 216-217: “Application of stylet to 216 adjust the shape of the endotracheal tube is incapable.” Please rephrase this sentence.

Response 7:

As suggestion by the reviewer, we rephrased the sentence to “Performers were not capable of adjusting the shape of the endotracheal tubes by application of stylets.” (Page 10 line 252-253)

Page 8 line 240-241: Although 1st attempt success rate was statistically significant between the groups; the clinical significance is less obvious with an OR of 1.06. This should be highlighted to the readers. If the study by Zhu et al is excluded, I suspect the statistical significance would also no longer be present.

Response 8:

As suggestion by both reviewers, we excluded Zhu et al in our study because it is conducted on patients with predicted difficult airway and therefore it cannot be pooled with the other 5 studies carried out in patients with normal airway. After we excluded the study, 1st attempt success rate was no longer statistically significant between the groups as the reviewer expected. We also renewed all the information about 1st attempt rate in the article.

Page 8 line 257-259: “Furthermore, one trial gave the opposite suggestions that Macintosh 257 laryngoscope provided shorter nasotracheal intubation times, better tracheal navigation, 258 requiring less use of the cuff inflation method.” Which study is this? The reference should be given.

Response 9:

We have added the indication of the reference that discussed about in pediatric population, Macintosh laryngoscope provided shorter nasotracheal intubation times, better tracheal navigation, and requiring less use of the cuff inflation method.

Reviewer 2 Report

Thank you for giving me the opportunity to contribute at the revision of this Systematic Review and meta-analysis on videolaryngoscope for nasotracheal intubation in oral and maxillofacial surgery_ A comparison of McGrath Videolaryngoscope versus Macintosh Laryngoscope for Nasotracheal Intubation: A Systematic Review and Meta-analysis.

I appreciate the effort of the authors, but the manuscript need to be revised in several areas in order to improve the quality.

Major suggestions:

  1. Could the authors formulate the review question, according to PICO template? The authors could use this example: In adult patients who underwent nasotracheal intubation for oral and maxillofacial surgery (P), does the use of McGrath Videolaryngoscope (I) compared to Macintosh Laryngoscope (C) increase first-attempt success rate, reduce intubation time, and improve glottis visualization (O)?”

  1. The authors should add in their manuscript the GRADE information. The certainty of evidence could be independently assessed by two authors based on the presence of limitations (risk of bias), indirectness, inconsistency, and imprecision according to the Grading of Recommendations Assessment, Development and Evaluation (GRADE) criteria.

  1. I disagree with inclusion of Zhu in the meta-analysis because it is conducted on patients with predicted difficult airway, so those results cannot be pooled with the other 5 studies carried out in patients with normal airway. Also, this study did not report some of the review findings, such as the rate of the Magill forceps used, and the rate of external laryngeal pressure maneuver used.

  1. I suggest revising the entire manuscript and results according to these considerations.

  1. The introduction should be improved by better explaining the rationale for this study, and what this study adds to the current literature.

  1. Discussion should be improved in contents and style.

  1. Please, add in supplementary materials a table showing the excluded studies (authors, title and journal) and the reasons for exclusion for each paper.

Minor suggestions:

ABSTRACT

Line 17: please, delete “..the outcomes, including…” and write directly the outcomes measured.

Line 17: add “…glottis visualization as rate of Cormack-Lehane classification 1, …”

Line 22-23: secondary outcomes are listed in the methods section. Please, move the sentence “The secondary outcomes were the rate of the Magill forceps used, and the external laryngeal pressure maneuver (BURP maneuver) used” at the end of background, after primary outcome.

Line 23-24: please, delete “and contained enough statistic data”, because it is a fundamental criterion for inclusion in meta-analysis. The authors could modify as follows: “Six articles met inclusion criteria and were included in the final analysis (n= 392 patients).

Line 26-27: “The rate of the Magill forceps used, and the external laryngeal pressure maneuver used were also found in the pooled analysis”. This sentence shoul be moved in methods, or in this section the results of rate of the Magill forceps and the external laryngeal pressure manoeuvre should be write.

INTRODUCTION:

Line 48: change “tracheal intubation” with “oro-tracheal intubation”.

Line 58: there a “dot” between “scope” and “with”. Please, delete it.

Line 57-62: the two sentences should be rewritten more clearly, because the assumptions of this systematic literature review are based on this concept.

Line 64-67: delete “the outcomes, including” and write directly what are the outcomes measured.

METHODS

Line 75-77: use the term “search string” and not the “keywords”. Please add the date of last search.

Line 77-79: I suggest changing  the sentence “Studies that have not been fully published or studies without full-text…” with “Reviews, case reports and studies published in abstract form were excluded”.

Line 86-88: Please add among exclusion criteria, studies involving nasotracheal intubation with any other videolaringoscope.

Line 127-128: delete the sentence “If at least 10 trials are identified, we will use funnel plots to determine publication bias”, or write in the results that this is not done and why.

RESULTS:

Line 141: please delete the first sentence, and insert reference to Fig 1 at the end of paragraph. 

Line 151: change the sentence as follows: “Eventually, 6 articles met our inclusion criteria and were included in the final analysis”

Line 155: please change “selected studies” with “included studies”.

Line 189-194: as the authors stated, the external laryngeal pressure maneuver was compared in 4 studies. So, please, delete Kwak in figure 6.

TABLE 1: I suggest to remove Zhu 2019, and include only 5 studies in the meta-analysis. Please, remove then the column of airway status.

FIGURE 1: please delete the term of “meta-analysis” in the figure legend. This is the whole process of search strategy that does not necessarily lead to a meta-analysis.

Authors should report the excluded records according to PICOS criteria. Example for population (pediatric, or not oral and maxillofacial surgery, etc…), for intervention (other VL), for comparison (not comparison with Macintosh Laryngoscope), for outcome or for study (non RCT).

ALL FOREST PLOTS: please, change “MgGrath” with “McGrath”

Fig 4. Forest plot of the first-attempt success rate shows Risk Ratio and not the Odds Ratio. Please, change it or change the text in the Data Synthesis, accordingly.

The inclusion of Zhu in this forest plot is in favour to McGrath, however, the results of this study revealed that McGrath was slightly superior to Macintosh in first attempt rate, but this trial included patients with predicted difficult intubations, so the studies are not comparable. Without Zhu, this difference probably could be even smaller.

Author Response

We thank the reviewer for the critical comments and constructive suggestions.

Thank you for giving me the opportunity to contribute at the revision of this Systematic Review and meta-analysis on videolaryngoscope for nasotracheal intubation in oral and maxillofacial surgery_ A comparison of McGrath Videolaryngoscope versus Macintosh Laryngoscope for Nasotracheal Intubation: A Systematic Review and Meta-analysis.

I appreciate the effort of the authors, but the manuscript need to be revised in several areas in order to improve the quality.

Major suggestions:

Could the authors formulate the review question, according to PICO template? The authors could use this example: In adult patients who underwent nasotracheal intubation for oral and maxillofacial surgery (P), does the use of McGrath Videolaryngoscope (I) compared to Macintosh Laryngoscope (C) increase first-attempt success rate, reduce intubation time, and improve glottis visualization (O)?”

Response 1:

We agreed with the importance to formulate the review question, and added PICO in introduction of our article.

The authors should add in their manuscript the GRADE information. The certainty of evidence could be independently assessed by two authors based on the presence of limitations (risk of bias), indirectness, inconsistency, and imprecision according to the Grading of Recommendations Assessment, Development and Evaluation (GRADE) criteria.

Response 2:

We also added the methodology and the result of assessments of GRADE to improve the quality our article.

I disagree with inclusion of Zhu in the meta-analysis because it is conducted on patients with predicted difficult airway, so those results cannot be pooled with the other 5 studies carried out in patients with normal airway. Also, this study did not report some of the review findings, such as the rate of the Magill forceps used, and the rate of external laryngeal pressure maneuver used.

Response 3:

As suggestion by the reviewer, we excluded Zhu et al in our review.

I suggest revising the entire manuscript and results according to these considerations.

Response 4:

Because we excluded Zhu et al in our review, we revised the entire article to strengthen our point of view.

The introduction should be improved by better explaining the rationale for this study, and what this study adds to the current literature.

Response 5:

We have revised our article entirely and explained the rationale of our study.

Discussion should be improved in contents and style.

Response 6:

The whole article was revised according to reviewer’s suggestions, and also edited by MDPI language editing services.

Please, add in supplementary materials a table showing the excluded studies (authors, title and journal) and the reasons for exclusion for each paper.

Response 7:

We have added the information of the studies that we excluded, and we also indicated why we excluded these studies. (1 predicted difficult airway, inadequate outcomes data, 1 pediatric population, and 5 other videolaryngoscope)

Minor suggestions:

ABSTRACT

Line 17: please, delete “..the outcomes, including…” and write directly the outcomes measured.

Response 8:

We revised the whole sentence in the introduction, and applied PICO template to reinforce the point of this article.

Line 17: add “…glottis visualization as rate of Cormack-Lehane classification 1, …”

Response 9:

After using the PICO template, we also added the details of glottis visualization.

Line 22-23: secondary outcomes are listed in the methods section. Please, move the sentence “The secondary outcomes were the rate of the Magill forceps used, and the external laryngeal pressure maneuver (BURP maneuver) used” at the end of background, after primary outcome.

Response 10:

As suggestion by the reviewer, we moved the statement to the end of background of the abstract

Line 23-24: please, delete “and contained enough statistic data”, because it is a fundamental criterion for inclusion in meta-analysis. The authors could modify as follows: “Six articles met inclusion criteria and were included in the final analysis (n= 392 patients).

Response 11:

As suggestion by the reviewer, we rephrased the sentence to “Five articles met inclusion criteria and were included in the final analysis (n= 331 patients).”

Line 26-27: “The rate of the Magill forceps used, and the external laryngeal pressure maneuver used were also found in the pooled analysis”. This sentence should be moved in methods, or in this section the results of rate of the Magill forceps and the external laryngeal pressure maneuver should be write.

Response 12:

We added the result of the secondary outcomes in this section. “Decreases in the rate of the Magill forceps used, and the external laryngeal pressure maneuver used were also found in the pooled analysis.”

INTRODUCTION:

Line 48: change “tracheal intubation” with “oro-tracheal intubation”.

Response 13:

As suggestion by the reviewer, we changed “tracheal intubation” with “oro-tracheal intubation” to specify the orotracheal intubation technique here.

Line 58: there a “dot” between “scope” and “with”. Please, delete it.

Response 14:

We deleted the dot here.

Line 57-62: the two sentences should be rewritten more clearly, because the assumptions of this systematic literature review are based on this concept.

Response 15:

As suggestion by the reviewer, we rewrote the two sentences.

Line 64-67: delete “the outcomes, including” and write directly what are the outcomes measured.

Response 16:

As suggestion by the reviewer, we rewrote the whole sentence here and used PICO to reinforce the point of this article.

METHODS

Line 75-77: use the term “search string” and not the “keywords”. Please add the date of last search.

Response 17:

We changed “keywords” with “search string”, and also added the last search date. (Page2, Line 82-84)

Line 77-79: I suggest changing the sentence “Studies that have not been fully published or studies without full-text…” with “Reviews, case reports and studies published in abstract form were excluded”.

Response 18:

As suggestion by the reviewer, we have replaced the sentence with “Reviews, case reports and studies published in abstract form were excluded”.

Line 86-88: Please add among exclusion criteria, studies involving nasotracheal intubation with any other videolaringoscope.

Response 19:

We added “the articles involving nasotracheal intubation with other videolaryngoscopes” at the end of exclusion criteria.

Line 127-128: delete the sentence “If at least 10 trials are identified, we will use funnel plots to determine publication bias”, or write in the results that this is not done and why.

Response 20:

As suggestion by the reviewer, we just included 5 articles in our review, so this sentence has been deleted.

RESULTS:

Line 141: please delete the first sentence, and insert reference to Fig 1 at the end of paragraph.

Response 21:

The first sentence was deleted and we added reference in Fig 1 at the end of this section.

Line 151: change the sentence as follows: “Eventually, 6 articles met our inclusion criteria and were included in the final analysis”

Response 22:

The sentence was change to “Eventually, 5 articles met our inclusion criteria and were included in the final analysis (n= 331 patients).”

Line 155: please change “selected studies” with “included studies”.

Response 23:

As suggestion by the reviewer, we changed the title with included studies (Line 185).

Line 189-194: as the authors stated, the external laryngeal pressure maneuver was compared in 4 studies. So, please, delete Kwak in figure 6.

Response 24:

We renewed the figure 6 that Kwak et al was deleted.

TABLE 1: I suggest to remove Zhu 2019, and include only 5 studies in the meta-analysis. Please, remove then the column of airway status.

Response 25:

As suggestion by the reviewer, we excluded Zhu et al in our analysis. We also renewed the Table 1 and stated the reason why we excluded the study in Figure 1.

FIGURE 1: please delete the term of “meta-analysis” in the figure legend. This is the whole process of search strategy that does not necessarily lead to a meta-analysis.

Authors should report the excluded records according to PICOS criteria. Example for population (pediatric, or not oral and maxillofacial surgery, etc…), for intervention (other VL), for comparison (not comparison with Macintosh Laryngoscope), for outcome or for study (non RCT).

Response 26:

As suggestion by the reviewer, meta-analysis was removed from the figure legend and some details were added in figure 1.

ALL FOREST PLOTS: please, change “MgGrath” with “McGrath”

Response 27:

We have corrected the spelling error in all figures.

Fig 4. Forest plot of the first-attempt success rate shows Risk Ratio and not the Odds Ratio. Please, change it or change the text in the Data Synthesis, accordingly.

Response 28:

We have added risk ratio in Data synthesis.

The inclusion of Zhu in this forest plot is in favour to McGrath, however, the results of this study revealed that McGrath was slightly superior to Macintosh in first attempt rate, but this trial included patients with predicted difficult intubations, so the studies are not comparable. Without Zhu, this difference probably could be even smaller.

Response 29:

As the reviewer stated, if we excluded Zhu et al in our analysis, the first attempt rate was no longer statistically significant. Because Zhu et al included the patients with predicted difficult airway, so we removed the data in our analysis. So the result and discussion were re-written in our article.

The manuscript is edited by MDPI language editing services.

Round 2

Reviewer 2 Report

The authors made the changes to the manuscript according my requeststhe authors made the changes to the manuscript according my requests.